# Study on SBS Optimal Block Ratio Based on Molecular Simulation

**DOI:** 10.3390/polym14224894

**Published:** 2022-11-13

**Authors:** Wenyue Liu, Chuanfeng Zheng, Haisong Luo, Xue Yang, Zhi Lin

**Affiliations:** 1College of Construction Engineering, Jilin University, Changchun 130061, China; 2College of Transportation, Jilin University, Changchun 130022, China

**Keywords:** SBS, block ratio, concentration profile, diffusion coefficient, adhesion work

## Abstract

The block ratio of SBS is an important factor influencing the swelling effect of modified asphalt, but the effect of the block ratio in the swelling process cannot be accurately studied by macro testing. In order to solve this problem and screen out the optimal SBS block ratio, molecular models of asphalt and SBS with different block ratios were established by molecular simulation technology at the microscopic level, and an asphalt–SBS interaction layer system was established on this basis. The diffusion and adhesion effects of SBS with different block ratios were evaluated by calculating the immersion depth, diffusion coefficient, and adhesion work of SBS in asphalt. The results show that SBS has a physical cross-linking reaction with asphalt during swelling, and SBS with a higher butadiene block ratio exhibits a deeper immersion depth; thus, SBS with a 3/7 block ratio has the best performance in the modification process, which is superior to SBS with other block ratios, in terms of both the diffusion and adhesion effect. The performance of asphalt modified by SBS with different block ratios was tested using penetration, softening point, and ductility, and the results were consistent with the simulation results, which proved the reliability of the microscopic conclusions from a macro perspective.

## 1. Introduction

The SBS modified asphalt mixture is one of the most widely used materials in China’s road paving industry. It has the characteristics of excellent high and low temperature performance and strong fatigue and water damage resistance [1,2,3]. Its main material, SBS modified asphalt, is prepared by blending styrene-butadiene-styrene polymer (SBS) with asphalt. Styrene block (PS block) has a high hardness, which can improve the high temperature performance of asphalt, and butadiene can improve the low temperature crack resistance of asphalt because of its strong elasticity and toughness [4,5,6]. There are many factors that affect the performance of SBS modified asphalt, such as SBS type, particle size, asphalt grade, mixing temperature, transportation time, etc. [7,8].

In recent years, researchers have focused on using various modifiers to improve the performance of SBS modified asphalt, hoping to balance the relationship between the cost of SBS modified asphalt and project quality [9,10,11,12]. Some researchers are devoted to solving the problem of the storage stability of SBS modified asphalt and have developed many stabilizers and compatibilizers that promote the mutual solubility of SBS and asphalt, such as sulfur powder and furfural extraction oil, which improve the problem of the segregation of the SBS modified asphalt to a certain extent, they make the asphalt too hard and brittle at low temperatures, resulting in poor low temperature performance [13,14,15,16]. Some researchers hope to solve the aging problem of modified asphalt by introducing a light component such as rejuvenating agents and antioxidants, but this often reduces the high temperature stability of asphalt [17,18,19]. These additives not only fail to fundamentally balance the high and low temperature performance of SBS modified asphalt, but may also affect the lifespan of modified asphalt, increase the cost, and pollute the environment.

Therefore, it is very important to study the influence of SBS’s own structure on the modification effect based on the physical properties of the modifier itself. In this paper, the asphalt and modifier model are established through Materials Studio 2019, and the properties of modified asphalt with different block ratios of SBS are studied at the microscopic level. The SBS with the optimal block ratio is screened out to develop higher performance SBS modifiers in the future. The asphalt test is used to verify the conclusions obtained through micro simulation.

## 2. Materials and Methods

In this paper, the interaction of asphalt and SBS in the swelling process was studied by establishing molecular models of asphalt and SBS. The modification mechanism of SBS was qualitatively analyzed by the apparent characteristics of the model after molecular dynamics simulation, and the optimal block ratio of SBS was explored by the quantitative analysis of the calculation results of molecular simulation. Molecular dynamics simulation is a microscopic simulation method based on Newton’s second law, which has a wide range of applications in the field of materials science.

### 2.1. Asphalt Model Establishment

In this paper, the 4-component 12-molecule model of Li and Greenfield is used, which has been proved to be able to simulate the physical and chemical properties of asphalt, as shown in Figure 1. The asphalt 12 molecule was assembled into matrix asphalt A1 by the amorphous cell module [20], as shown in Figure 2. The results of the four-component ratio were obtained according to the component separation test, as shown in Table 1.

The established asphalt model is optimized for 100,000 steps in the COMPASS force field to minimize the energy of the system and obtain the most stable structure of the asphalt. In order to release the energy generated in the formation and optimization of the model, use the “anneal” option in the Forcite module to anneal the model; the annealing temperature is 300 K~500 K, the ensemble uses NVT, and the number of steps is 25,000. The asphalt model, before and after optimization, is shown in Figure 2.

### 2.2. Verification of the Rationality of the Asphalt Model

Since asphalt is a complex polymer, the reliability of the asphalt model is verified by comparing the simulated density with the measured density. After geometric optimization and annealing of the established matrix asphalt model, molecular dynamics calculations were performed on the model under the NPT ensemble. The energy and density changes in the entire system during the simulation process are shown in Figure 3 and Figure 4. It can be seen from Figure 3 that the kinetic energy and potential energy in the system reach equilibrium at about 20 ps, which means that the system is basically stable at this time. The density in Figure 4 is basically stable after 20 ps. The density value after the simulation can represent the density value of the asphalt model at steady state. Finally, the densities of the two groups of models were obtained as 1.016 g/cm^3^ and 1.028 g/cm^3^, respectively. The measured densities of the two asphalts are 1.053 g/cm^3^ and 1.072 g/cm^3^, respectively, and the errors between the model and the measured sample are 1.6% and 4.1%, respectively. This error is small, so the model can essentially represent the measured sample [21].

The compatibility of polymers is also an important indicator for the rationality verification of the model. The solubility parameter, as a physical parameter to evaluate the degree of compatibility between different polymer molecules, can be used as the basis for the verification of the molecular compatibility of each component in asphalt. For the solubility parameter (SP) of 12 asphalt molecules, the calculation formula is shown in Equation (1), and the obtained data is shown in Table 2.
(1)$$\mathrm{SP}=\sqrt{E/V}$$

In the formula: SP is the solubility parameter; E is the cohesive energy; V is the volume; and E/V is the cohesive energy density.

It can be seen from Table 2 that the maximum difference of the SP of each component of the matrix asphalt model is 0.75 (J/cm^3^)^1/2^, which is less than 4 (J/cm^3^)^1/2^. Therefore, the compatibility of each component of the matrix asphalt is good, and the model is relatively stable. Therefore, the model established above can represent the actual asphalt.

### 2.3. SBS Model Establishment

SBS is a block copolymer formed by the polymerization of styrene and butadiene; the molecular structures of styrene, butadiene, and SBS are shown in Figure 5a, Figure 5b, and Figure 5c, respectively. In Figure 5c, n is the number of styrene monomers in the polystyrene segment (PS), and m is the number of butadiene monomers in the butadiene segment (PB). The linear SBS molecular model was drawn according to the block ratios of 2/8, 3/7, 4/6, and 5/5 by the Materials Studio 2019 software, as shown in Figure 6, and 10,000 steps of geometric optimization were carried out. The block ratios of common SBS on the market are 2/8, 3/7, and 4/6, and 5/5 is the envisaged block ratio. The basic properties of SBS with four different block ratios at the micro level were obtained through molecular simulation, including glass transition temperature (Tg), density, CED, SP, and micro viscosity. Except for the glass transition temperature simulation, all other parameters were 170 °C, as shown in Table 3.

### 2.4. Establishment of SBS–Asphalt Interaction Model

The optimized SBS molecular model and asphalt model were assembled into an SBS–asphalt layer model using the “build layer” option, in which a vacuum layer with a thickness of 60 Å was set above the SBS layer in order to avoid the influence of the periodic structure on the simulation. In order to ensure that the established layer model is in the lowest energy state before the dynamics simulation, 100,000 steps of geometric optimization and 25,000 steps of annealing simulation are performed on the layer model under the NVT ensemble. The final model is shown in Figure 7.

### 2.5. Molecular Dynamics

Using the “dynamic” option in the Forcite module, the above layer model was simulated under the NPT ensemble for 100 ps, and the simulation step was 1 fs. The output was analyzed to study the diffusion effect of SBS in the asphalt and the adhesion effect at the interactive interface.

### 2.6. Asphalt Test

Penetration, softening point, and ductility were used to explore the modification effect of SBS with different block ratios on asphalt. The content of SBS is 6%. Maoming 70 # asphalt was used as the base asphalt. The performance parameters of SBS and base asphalt are shown in Table 4 and Table 5. The consistency of SBS modified asphalt was measured by penetration; the temperature sensitivity and consistency of SBS modified asphalt were measured by the softening point; the resistance to plastic deformation of SBS modified asphalt was measured by ductility.

## 3. Results and Discussion

The qualitative analysis of the interaction effect between SBS with different block ratios and asphalt was carried out through the output images of dynamic simulation. The wetting depth of SBS on the asphalt surface is quantitatively analyzed by the concentration profile curve of SBS in the Z direction, and the interaction rate of SBS in asphalt is quantitatively evaluated by the mean square displacement (MSD) curve of SBS. The attraction between SBS and asphalt is quantitatively analyzed by the adhesion work between SBS and asphalt.

### 3.1. Image Analysis after Dynamics Simulation

The SBS–asphalt interaction model after dynamics simulation is shown in Figure 8. As shown in the figure, after dynamic simulation, the SBS molecules intertwine with the asphalt molecules. During the interaction with asphalt, it can be observed that SBS with block ratios of 2/8, 3/7, and 4/6 has obvious interactions with aromatics and saturates. This is consistent with the existing known mechanism by which SBS absorbs light components to form a stable network structure when it swells, which proves the rationality of the SBS and asphalt model. However, SBS with a block ratio of 5/5 accumulates on the asphalt surface, and the interaction effect is poor, which qualitatively indicates that the proposed SBS model with a block ratio of 5/5 exhibits shortcomings.

### 3.2. Analysis of the Diffusion Effect

The qualitative analysis of the simulated images cannot fully reflect the interaction between asphalt and SBS during the swelling process. Therefore, this paper quantitatively evaluates the effect of each SBS block ratio on the diffusion effect by analyzing the infiltration concentration profile and the MSD of the two. The infiltration of SBS in asphalt is calculated using the “concentration profile” option in Forcite to calculate the SBS concentration distribution in the Z direction. The results are shown in Figure 9.

It can be seen from Figure 9 that the SBS infiltration depth in asphalt is mainly in the range of 0~30 Å, but the SBS distribution is different for different block ratios. SBS with a block ratio of 2/8 has the most uniform distribution in asphalt, with a distribution range of 0 Å~25 Å. This is mainly due to the long chain-like structure, with a large degree of freedom in the butadiene block, which can easily enter into asphalt molecules. However, with the increasing proportion of styrene block, the distribution of SBS in asphalt is gradually concentrated, and the value of the relative distribution concentration at the peak value is gradually increased, indicating that the diffusion ability in asphalt is gradually reduced; thus, the ability to form a three-dimensional spatial network structure in asphalt is gradually weakened.

The interdiffusion rate between SBS and asphalt can be obtained through the MSD curve. MSD refers to the difference between an atom and the mean of the displacements of all atoms. The MSD of asphalt and SBS can represent the diffusion of the asphalt and SBS molecules during this period. The calculation method is shown in Equation (2).
(2)$$MSD\left(t\right)={\langle \left|r_{i}\left(t\right)-r_{i}\left(0\right)\right|}^{2}\rangle$$

In the formula: r_i_ (0) and r_i_ (t) are the displacement of the particle at the initial moment and at time t; 〈 〉 is the average of all atoms in the group.

For quantitative analysis, the mutual diffusion rate of the two is calculated by Formula (3); and the slope of the MSD can be directly obtained by dividing by 6.
(3)$$D=\frac{1}{6}\underset{t\to \infty }{\lim}\frac{d}{dt}\overset{N}{\underset{i=1}{\sum }}{\langle \left|r_{i}\left(t\right)-r_{i}\left(0\right)\right|}^{2}\rangle$$

The MSD of SBS and asphalt is shown in the process of mutual diffusion, as illustrated in Figure 9.

It can be seen from Figure 10a that the diffusion effect of SBS with different block ratios in the first 10 ps was not significantly different. After 10 ps, the influence of block ratio began to play a significant role. The diffusion coefficient of SBS is obtained by fitting and analyzing the mean square displacement curve according to Formula (3). The diffusion coefficients of 2/8, 3/7, 4/6 and 5/5 are 1.65608, 2.06852, 1.44826, and 1.86168 respectively. The diffusion rate in asphalt is ranked from strong to weak, and 3/7 > 5/5 is obtained > 2/8 > 4/6; this result shows that the block SBS has more advantages than the SBS with 3/7 in the process of thermal movement with asphalt.

It can be seen from Figure 10b that the diffusion effect of asphalt in SBS does not change significantly with the block ratio of SBS. The main reason is that the volume of asphalt is larger than that of SBS, and the amount of diffusion into SBS is too small when calculating the MSD, so it cannot be used to prepare to characterize the swelling process. Therefore, the optimal way to evaluate the diffusion effect of SBS and asphalt in the system is to refer to the MSD of SBS diffusion in asphalt.

### 3.3. Analysis of the Adhesion Effect

Microscopically, the adhesive performance between interfaces is evaluated by the adhesion work, and the adhesion work is defined as the energy required to separate the asphalt from the SBS. The specific method is shown in Formula (4). The adhesion work between asphalt and SBS is mainly composed of non-bond energy, which includes van der Waals force and the Coulomb electrostatic force. The van der Waals force and the Coulomb electrostatic force between SBS and asphalt can be calculated after dynamic simulation, and the total adhesion work can be obtained through Formulas (4) and (5). Specific results are shown in Table 6.
(4)$$\Delta E=E_{Aspahlt}+E_{SBS}-E_{total}$$

In the formula: Δ*E* is the adhesion work between asphalt and *SBS* (kcal/mol); *E_Aspahlt_* and *E*_SBS_ are the energy of single asphalt and single SBS (kcal/mol); and *E*_total_ is the total energy of the asphalt and SBS (kcal/mol).
(5)$$\Delta E_{adhesion}=\Delta E_{\mathrm{van}\;\mathrm{der}\;\mathrm{Waals}}+\Delta E_{\mathrm{Electrostatic}}$$

In the formula: Δ*E_adhesion_* is the total adhesion work (kcal/mol); Δ*E*_van der Waals_ and *ΔE*_Electrostatic_ are the part of adhesion work between two substances due to the interaction of the van der Waals force and the Coulomb electrostatic force, respectively (kcal/mol).

It can be seen from Table 6 that the adhesion work generated by the van der Waals force of SBS and asphalt is dominant in the total adhesion work, and the adhesion work generated by the electric field force is negligible. Therefore, it can be concluded that the adhesion effect between SBS and asphalt is mainly caused by the van der Waals force. When the total adhesion work is negative, it indicates that attraction is generated between two substances, and when the total adhesion work is positive, it indicates that repulsion is generated between two substances. The greater the absolute value of attraction and repulsion, the stronger the attraction or repulsion effect. According to the results in Table 4, SBS with a block ratio of 2/8 and 3/7 shows attraction with asphalt, and the attraction between SBS with a block ratio of 3/7 and asphalt is stronger than that of SBS with a block ratio of 2/8. From the perspective of enhancing the bonding effect between the two substances, SBS with a block ratio of 3/7 has the most advantages. The repulsive force between SBS and asphalt with block ratio of 4/6 and 5/5 is not conducive to the interaction between SBS and asphalt.

### 3.4. Asphalt Test Analysis

The penetration, softening point, and ductility test results of SBS modified asphalt under different block ratios are shown in Table 7.

According to the results for the penetration, softening point, and ductility of asphalt modified by SBS with different block ratios, the SBS block ratio has an impact on the performance of asphalt, to a certain extent. According to the test results for penetration, the consistency of SBS modified asphalt increases with the increase in styrene content in SBS. This is because the increase in hard segment content in SBS improves the overall strength of SBS modified asphalt. It can be seen from the test results for softening point that the order of softening point from large to small is 3/7 > 2/8 > 4/6, and the softening point of asphalt modified by SBS with a block ratio of 3/7 is the highest because the interaction between asphalt modified by SBS with a block ratio of 3/7 is the best, resulting in a better effect regarding resisting temperature sensitivity. The order of ductility of asphalt modified by SBS with different block ratios is 3/7 > 2/8 > 4/6 from large to small, which indicates that asphalt modified by SBS with a block ratio of 3/7 has the best resistance to plastic deformation at 15 °C. According to the test results for the three indicators, the test results obtained through molecular simulation are reliable.

## 4. Conclusions

In this paper, by establishing molecular models of asphalt and SBS with different block ratios, and assembling the two into an interactive model, the diffusion effect and bonding effect of SBS with different block ratios in the swelling process were studied, and the following conclusions were obtained:(1)It can be seen from the image output after the dynamics simulation that SBS can entangle and interpenetrate with asphalt, to some extent, after the molecular dynamics simulation, but the interaction between SBS with different block ratios and asphalt is obviously different. SBS with block ratios of 2/8, 3/7, and 4/6 have good interaction effects with asphalt, which is consistent with the existing known SBS swelling mechanism, while SBS with block ratios of 5/5 accumulate on the asphalt surface, and the interaction effect is poor.(2)In terms of the quantitative evaluation of the diffusion effect, the diffusion ability of SBS was evaluated by the concentration distribution of SBS in the Z direction, and the order of diffusion ability of SBS with different block ratios was 2/8 >3/7 > 4/6 > 5/5. The diffusion rate of SBS in asphalt was evaluated by the diffusion coefficient, and the order of the diffusion rate of SBS with different block ratios was 3/7 >2/8 > 4/6 > 5/5.(3)In terms of the adhesion effect, the adhesion work between SBS and asphalt with different block ratios was evaluated, and it was found that SBS with block ratios of 2/8 and 3/7 showed an attraction to asphalt, and the adhesion between SBS with a block ratio of 3/7 and asphalt was twice that of the 2/8 block ratio. SBS with block ratios of 4/6 and 5/5 have repulsive force with asphalt; thus, SBS with a block ratio of 3/7 shows the best adhesion effect.(4)The penetration of asphalt modified by SBS with different block ratios is 2/8 > 3/7 > 4/6, from large to small, and the softening point and ductility are 3/7 > 2/8 > 4/6, from strong to weak. The test results prove the reliability of the conclusions obtained by molecular simulation. From the consistency, temperature sensitivity, and the ability of the asphalt to resist plastic deformation, it is proved that the asphalt modified by SBS with a block ratio of 3/7 has the best performance, which is consistent with the results obtained by simulation.

In summary, SBS with a block ratio of 3/7 is better than the other three ratios in terms of diffusion and adhesion effect, so 3/7 is the optimal block ratio, and the asphalt test proves that the microscopic conclusion is reliable.

## Figures and Tables

**Figure 1 polymers-14-04894-f001:**
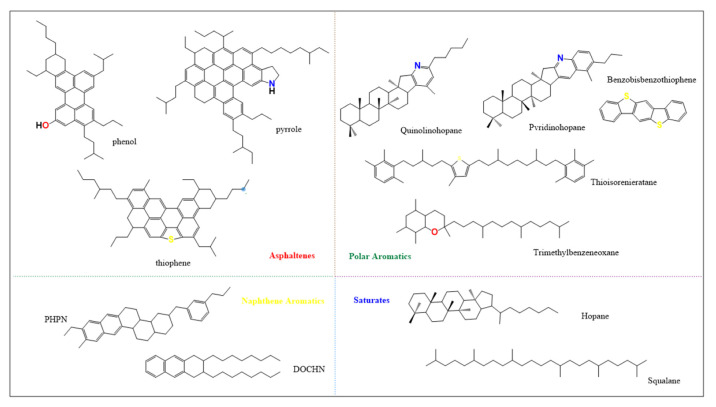
The four-component 12-molecule model of asphalt.

**Figure 2 polymers-14-04894-f002:**
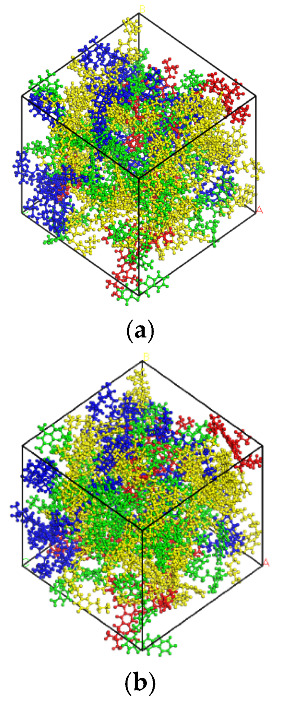
Molecular model of matrix asphalt before and after optimization. (**a**) Before optimization. (**b**) After optimization.

**Figure 3 polymers-14-04894-f003:**
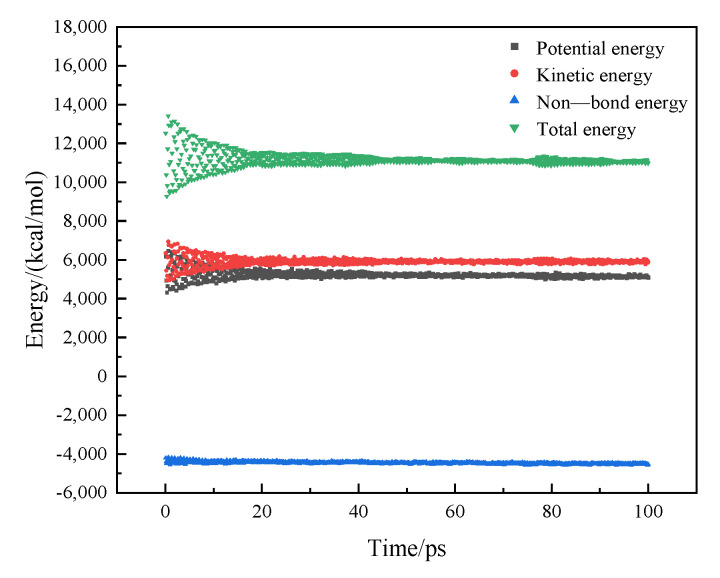
The energy of the asphalt model during the simulation.

**Figure 4 polymers-14-04894-f004:**
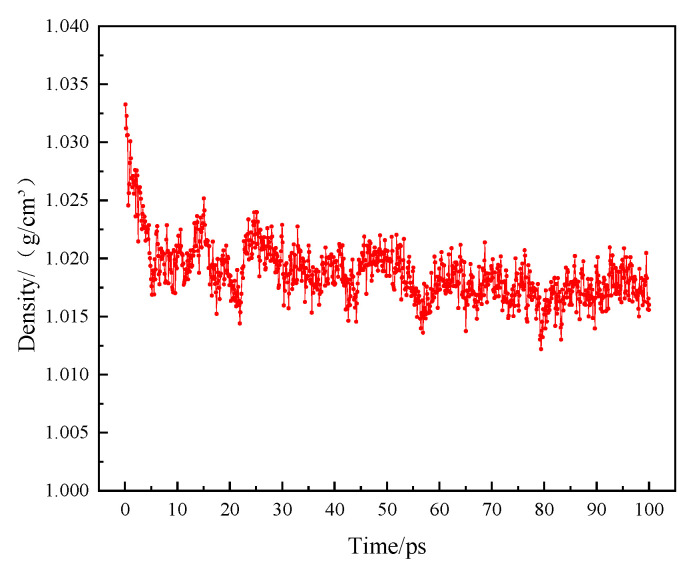
The density of the asphalt model during simulation.

**Figure 5 polymers-14-04894-f005:**
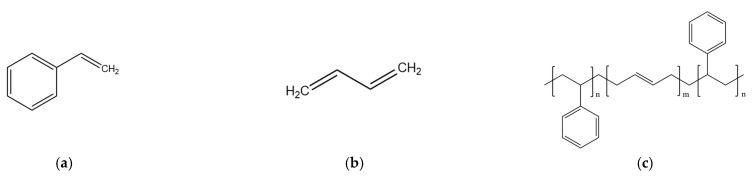
Molecular Structure. (**a**) Styrene. (**b**) Butadiene. (**c**) SBS.

**Figure 6 polymers-14-04894-f006:**
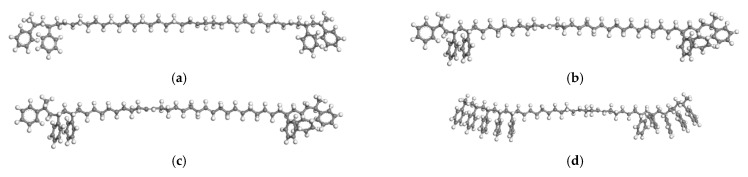
SBS molecular models with different block ratios. (**a**) 2/8. (**b**) 3/7. (**c**) 4/6. (**d**) 5/5.

**Figure 7 polymers-14-04894-f007:**
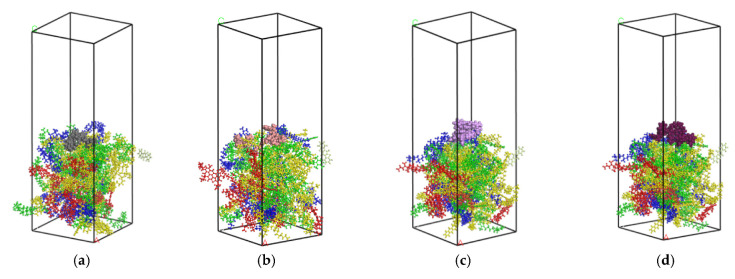
SBS–asphalt interaction model. (**a**) 2/8. (**b**) 3/7. (**c**) 4/6. (**d**) 5/5.

**Figure 8 polymers-14-04894-f008:**
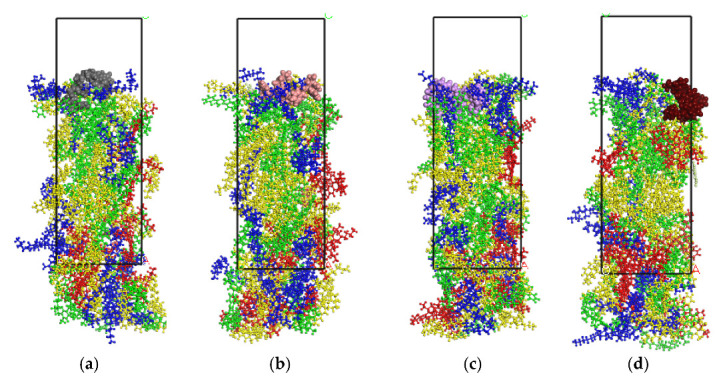
After molecular dynamics simulation of the asphalt–SBS interaction layer. (**a**) 2/8. (**b**) 3/7. (**c**) 4/6. (**d**) 5/5.

**Figure 9 polymers-14-04894-f009:**
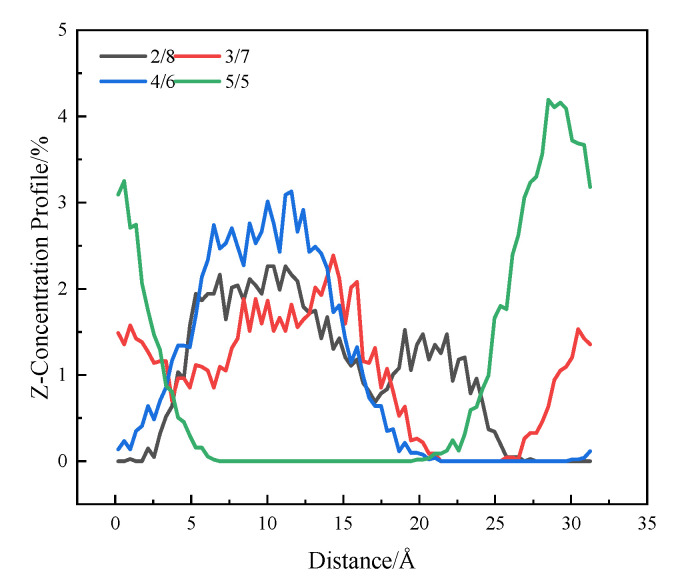
Wetting of SBS in asphalt.

**Figure 10 polymers-14-04894-f010:**
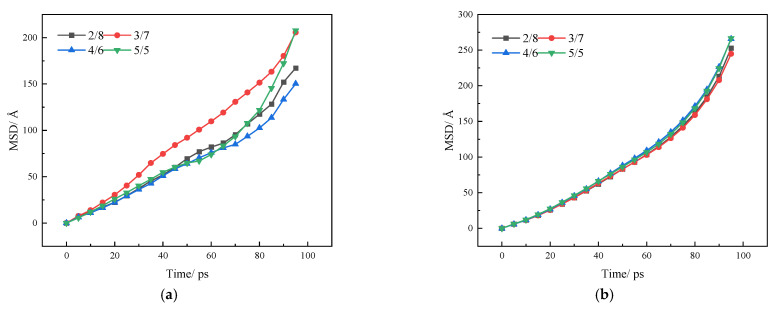
MSD of mutual diffusion between asphalt and SBS. (**a**) MSD of SBS diffusion in asphalt, (**b**) MSD of asphalt diffusion in SBS.

**Table 1 polymers-14-04894-t001:** The four-component mass fraction.

Asphalt Component		Number of Molecules	Mass Fraction/%
Asphaltene	Asphaltene-phenol	3	13.3
	Asphaltene-pyrrole	3	
	Asphaltene-thiophene	2	
Polar Aromatics	Pyridinohopane	4	31.1
	Thin-isorenieratane	7	
	Benzobisbenzothiophene	6	
	Quinolinohopane	4	
	Trimethylbenzene-oxane	7	
Naphthene Aromatics	PHPN	17	37.6
	DOCHN	18	
Saturates	Squalene	7	18.0
	Hopane	8	

**Table 2 polymers-14-04894-t002:** The cohesive energy density and solubility parameters of the asphalt components.

Asphalt Component		Cohesive Energy Density/(J/m^3^)	Solubility Parameter/(J/cm^3^)1/2
Asphaltene	Asphaltene-phenol	2.77 × 10^8^	16.64
	Asphaltene-pyrrole	2.71 × 10^8^	16.46
	Asphaltene-thiophene	2.65 × 10^8^	16.28
Polar Aromatics	Pyridinohopane	2.65 × 10^8^	16.28
	Thin-isorenieratane	2.78 × 10^8^	16.67
	Benzobisbenzothiophene	2.86 × 10^8^	16.91
	Quinolinohopane	2.71 × 10^8^	16.47
	Trimethylbenzene-oxane	2.72 × 10^8^	16.49
Naphthene Aromatics	PHPN	2.79 × 10^8^	16.70
	DOCHN	2.75 × 10^8^	16.58
Saturates	Squalene	2.62 × 10^8^	16.19
	Hopane	2.61 × 10^8^	16.16
Maximum difference value		0.25 × 10^8^	0.75

**Table 3 polymers-14-04894-t003:** Basic properties of SBS with different block ratios.

SBS Block Ratio	Density/ (g/cm^3^)	Cohesive Energy Density /(J/m^3^)	Solubility Parameter /(J/cm^3^)^1/2^	Micro Viscosity/cP	Glass Transition Temperature /°C	
					Tg1	Tg2
2/8	0.885	2.803716 × 10^8^	16.74429989	1.188	−79	71
3/7	0.904	2.645003 × 10^8^	16.26346573	1.576	−75	70
4/6	0.940	2.697376 × 10^8^	16.42368958	1.658	−73	72
5/5	0.960	2.669428 × 10^8^	16.33838464	2.007	−72	75

**Table 4 polymers-14-04894-t004:** Basic performance indexes of SBS with different block ratios.

SBS Block Ratio	Shore Hardness/A	300% Constant Elongation Stress/MPa	Tensile Strength/MPa	Elongation at Break/%	Ash Content/%
2/8	58	1.4	8.0	700	0.20
3/7	68	2.0	15.0	700	0.21
4/6	85	3.5	24.0	730	0.20

**Table 5 polymers-14-04894-t005:** Basic performance indexes of asphalt.

Asphalt	Penetration /0.1 mm	15 °C Ductility/cm	Softening Point/°C	60 °C Standard Viscosity/(Pa.s)
Maoming 70 #	73	106	48.3	5144

**Table 6 polymers-14-04894-t006:** Adhesion work of SBS and asphalt.

SBS Block Ratio	ΔE _van der Waals_	ΔE _Electrostatic_	ΔE _Total_
2/8	−109.332	6.882	−102.45
3/7	−212.772	5.573	−207.199
4/6	683.816	11.054	694.87
5/5	670.133	13.117	683.25

**Table 7 polymers-14-04894-t007:** Three indicators of asphalt modified by SBS with different block ratios.

SBS Block Ratio	Dosage/%	Penetration/0.1 mm	Softening Point/°C	15 °C Ductility/cm
2/8	6	46.7	86.5	46.62
3/7		46.2	87.9	47.69
4/6		44.6	85.4	45.92

## Data Availability

The date presented in the study are available on request from the corresponding author.

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
