# Peer review of "Study on SBS Optimal Block Ratio Based on Molecular Simulation"

_polymers, 2022, doi:10.3390/polym14224894_

Round 1

Reviewer 1 Report

A very specialized manuscript. State of the art review well done. Describe if anyone has studied the structure of SBS and asphalt and used a similar model. In the conclusions, provide practical application in road construction.

Author Response

Response to Reviewer 1 Comments

Point 1: A very specialized manuscript. State of the art review well done. Describe if anyone has studied the structure of SBS and asphalt and used a similar model. In the conclusions, provide practical application in road construction.

Response 1: Thank you very much for your valuable comments. At present, there are few molecular simulation studies on SBS and asphalt, especially on the influence of SBS block ratio on its modification mechanism. We will gradually deepen the research on the interaction between SBS molecules and asphalt molecules in the future research, and further deepen the understanding of the interaction mechanism between them at the micro level, providing a theoretical basis for the actual asphalt pavement.

Reviewer 2 Report

Thank you for submitting your work.

In my general opinion, the work describes an interesting idea to evaluate and study deeper the interaction of SBS and asphalt and which is the optimal SBS composition to reach the best interaction conditions. I would like to highlight some points to increase readers' understanding.

In the abstract, line 15 "...SBS has physical cross-linking", what do exactly you mean by physical cross-linking? To the best of my knowledge, It does not exist physical cross-linking.

1) In the introduction section:

i. In line 29 "...SBS type...", Could you explain better what would you mean by "SBS type"?

ii) In line 44, the authors mentioned the importance of SBS's structure. Then I believe It could be interesting to show explicitly its structure because the cross-linking reaction of SBS is due to the double bonds present in the butadiene structure. However, I think here it is not clear to the readers. And the authors have to make clear that there are just physical INTERACTIONS between SBS and asphalt as I could see at the end of the work.

iii) Could the authors explain more why they have used Materials studio 2019 as software?

2) In the materials and methods section:

i) Firstly, It is not important to inform the source of SBS? It was synthesized in the Lab or is a commercial SBS? Depending on the approach, we can reach different microstructures.

ii) The cross-linking reaction occurs mainly between butadiene and which molecule from asphalt?

iii) Following Table 2, I think the authors could add another table describing the general properties of the different SBS such as Tg as a function of SBS composition and structure.

iv) In the first equation, could the authors explain how they calculated the cohesive energy density? Or these are values from the literature? They have to insert this information below the table. Also, does the solubility parameter for SBS not change as a function of the block ratio?

v) In my opinion, before showing these 3D structures (very good as well), the authors should present a chemical structure to make more understandable these 3D structures.

3) Results and discussion

i. The explanation of Figure 8 seems to be not clear. In fact, the authors began the explanation by saying that the max infiltration depths are roughly the same. However, they explained just in line 172 that they adopted the integrating area under the curve.

ii. Could you explain better why the 2/8 is the deepest than the other three between 25 and 30? Taking into account the graph, I would say 5/5 or 3/7 in this range.

iii) In line 197, "...which are 1.65608, 2.06852..., respectively." I cannot attribute the diffusion coefficients to each block ratio. It is not clear to me. If I consider 2/8 (1.65608), 3/7 (2.06852), 4/6, and 5/5; Could the authors explain why the block ratio 4/6 presented the lowest coefficient ratio even compared to 2/8 and 5/5?

iv) In Table 4, the authors should provide from where they have energy values. (as subscript information below the table) Or If they have calculated, they should provide the formula used. In my point of view, the authors should give reshape the discussion of Table 4.

Author Response

Response to Reviewer 2 Comments

Point 1: Thank you for submitting your work. In my general opinion, the work describes an interesting idea to evaluate and study deeper the interaction of SBS and asphalt and which is the optimal SBS composition to reach the best interaction conditions. I would like to highlight some points to increase readers' understanding.

In the abstract, line 15 "...SBS has physical cross-linking", what do exactly you mean by physical cross-linking? To the best of my knowledge, It does not exist physical cross-linking.

Response 1: Thank you very much for your valuable comments. Physical cross-linking in this article refers to the interaction between SBS and saturates and naphthene aromatic in the process of SBS adsorbing asphalt light components. It may be that we have not explained it in this article, and we have added the description of relevant parts.

Point 2:

1) In the introduction section:

i). In line 29 "...SBS type...", Could you explain better what would you mean by "SBS type"?

  1. ii) In line 44, the authors mentioned the importance of SBS's structure. Then I believe It could be interesting to show explicitly its structure because the cross-linking reaction of SBS is due to the double bonds present in the butadiene structure. However, I think here it is not clear to the readers. And the authors have to make clear that there are just physical INTERACTIONS between SBS and asphalt as I could see at the end of the work.

iii) Could the authors explain more why they have used Materials studio 2019 as software?

Response 2: Thank you very much for your valuable comments.

i). According to the molecular structure, common SBS are classified into block linear structure and star structure. The relative molecular weight of block type SBS molecule is lower than that of star type SBS molecule, and it has better compatibility with asphalt. Star shaped SBS molecules have greater cohesive energy strength due to their relatively large molecular weight. Both are block copolymers formed by polymerization of styrene and butadiene, but their molecular configurations are different, so they have different "SBS types".

  1. ii) After careful consideration of the reviewer's comments, we found that the description of the interaction mechanism between asphalt and SBS molecules by molecular simulation in the original manuscript was not accurate enough. For this problem, we deleted "chemical". Please refer to line 42 for details.

iii) Materials Studio 2019 has both universal and specific force fields, and is applicable to various molecular models. It can quantitatively study the physical interaction between asphalt molecules and SBS molecules at the molecular level. The software has powerful pre-processing and post-processing functions. The pre-processing part can draw the molecular model through its own module, or import the molecular model through its own file library. In the post-processing process, specific data such as density, solubility parameters (SP), the mean square displacement (MSD) and concentration distribution can be directly output through the analysis module. Through the data that cannot be obtained from these macro experiments, researchers can more conveniently study the micro behavior of asphalt and SBS molecules. The updated version of Materials Studio 2019 was not used because the 2019 version of the software already has the functions required for this study.

Point 3:

2) In the materials and methods section:

  1. i) Firstly, It is not important to inform the source of SBS? It was synthesized in the Lab or is a commercial SBS? Depending on the approach, we can reach different microstructures.
  2. ii) The cross-linking reaction occurs mainly between butadiene and which molecule from asphalt?

iii) Following Table 2, I think the authors could add another table describing the general properties of the different SBS such as Tg as a function of SBS composition and structure.

  1. iv) In the first equation, could the authors explain how they calculated the cohesive energy density? Or these are values from the literature? They have to insert this information below the table. Also, does the solubility parameter for SBS not change as a function of the block ratio?
  2. v) In my opinion, before showing these 3D structures (very good as well), the authors should present a chemical structure to make more understandable these 3D structures.

Response 3: Thank you very much for your valuable comments.

  1. i) SBS with block ratio of 2/8, 3/7 and 4/6 are common linear SBS, and SBS with block ratio of 5/5 is established through software. After carefully considering the comments of the reviewer, we think it is very necessary to describe the source of SBS in detail in Section 2.3. The detailed changes are in lines 114 and 115
  2. ii) After carefully considering the comments of reviewers, we found that the cross-linking mechanism between SBS molecules and asphalt molecules in the original manuscript was not clear enough. Butadiene segment mainly reacts with light components (saturates and naphthene aromatic) in asphalt. Butadiene segment (PB segment) in SBS absorbs light components (saturates and naphthene aromatic) of asphalt under high temperature, causing the volume of PB segment to expand. The Styrene segment (PS segment) in SBS hardens again during the cooling process, forming a three-dimensional spatial network structure together with the butadiene segment that absorbs the light components of asphalt.

iii) According to your valuable suggestions, a table 3 describing the general microscopic properties of SBS with different block ratios is added after Table 2. Table 3 mainly includes the microscopic viscosity, density, solubility parameters at 170 ℃ and glass transition temperature of four SBS. The detailed modification is in Table 3 of Section 2.3.

  1. iv) The cohesive energy density (CED) of SBS model after dynamic simulation is calculated through the Forcite module, and the calculated CED is obtained in the output file. CED refers to the energy required by 1mol condensate in unit volume to overcome the intermolecular force when vaporizing. When the CED of two substances is closer, the compatibility is better.

The solubility parameters of SBS change with the change of block ratio. We supplement this part in Table 3 of Section 2.3.

  1. v) According to your valuable comments, we added the molecular structure of butadiene,styrene and SBS in Section 2.3 of the text. Please refer to the modified Figure 5 for details.

Point 4:

3) Results and discussion:

  1. i) The explanation of Figure 8 seems to be not clear. In fact, the authors began the explanation by saying that the max infiltration depths are roughly the same. However, they explained just in line 172 that they adopted the integrating area under the curve.
  2. ii) Could you explain better why the 2/8 is the deepest than the other three between 25 and 30? Taking into account the graph, I would say 5/5 or 3/7 in this range.

iii) In line 197, "...which are 1.65608, 2.06852..., respectively." I cannot attribute the diffusion coefficients to each block ratio. It is not clear to me. If I consider 2/8 (1.65608), 3/7 (2.06852), 4/6, and 5/5; Could the authors explain why the block ratio 4/6 presented the lowest coefficient ratio even compared to 2/8 and 5/5?

  1. iv) In Table 4, the authors should provide from where they have energy values. (as subscript information below the table) Or If they have calculated, they should provide the formula used. In my point of view, the authors should give reshape the discussion of Table 4

Response 4: Thank you very much for your valuable comments.

  1. i) After carefully considering the comments of the reviewers, we found that the explanations of Figure 8 was not very clear, so we discussed it again. For detailed modifications, please refer to Sections 3.1. Please refer to 152 lines - 160 lines for details.
  2. ii) After carefully considering the comments of the reviewers, we found that the explanations of Figure 9 was not very clear, so we discussed it again. For detailed modifications, please refer to Sections 3.2. Please refer to 173 lines - 182 lines for details.

iii) After carefully considering the comments of the reviewers, we find that the paragraph is due to the semantic ambiguity in the process of writing, and we have made specific modifications in the text. Please refer to 197 lines -204 lines for details.

  1. iv) The data obtained in Table 4 are from the output file of SBS-asphalt interaction model after dynamic simulation. After carefully considering the comments of reviewers, we added Formula 5 to improve the calculation process of adhesion work. We have reshaped the discussion of Table 4. Please refer to 229 lines -241 lines for details.

Round 2

Reviewer 2 Report

In general, the authors have answered my questions but I think the work is still fragile in terms of speculating a lot and not giving us experimental proofs to validate their ideas even being a theoretical study.

In my opinion, the work needs some additional experiments to validate their affirmations and ideas. 

Author Response

Response to Reviewer 2 Comments

Point 1:

In general, the authors have answered my questions but I think the work is still fragile in terms of speculating a lot and not giving us experimental proofs to validate their ideas even being a theoretical study.

In my opinion, the work needs some additional experiments to validate their affirmations and ideas.

Response 1:

Thank you very much for your valuable comments. At the macro level, three index tests of asphalt modified by SBS with different block ratios were added to verify the conclusions obtained from the micro molecular simulation, and the corresponding additions and modifications were made in the paper.

Among them, add relevant instructions of the test part in the summary, please see lines 18-20 for details;

Add relevant instructions of the test part in the foreword, please see lines 47-48 for details;

Section 2.6 is added, which mainly includes the preparation of experimental materials, such as the performance parameters of asphalt and SBS with different block ratios and test method adopted. For details, please refer to the revised lines 149-155, Table 4 and Table 5;

Add section 3.4 to describe the test results, please refer to lines 260-273 and Table 7 for details;

Add 4) to the conclusion part to improve the description of the test part. Please refer to lines 295-300 for details.

Round 3

Reviewer 2 Report

Following a number of modifications that make the work clearer, I suggest the acceptance of the work in the present form.